# An Outbreak in Pigeons Caused by the Subgenotype VI.2.1.2 of Newcastle Disease Virus in Brazil

**DOI:** 10.3390/v13122446

**Published:** 2021-12-06

**Authors:** Luciano M. Thomazelli, Juliana A. Sinhorini, Danielle B. L. Oliveira, Terezinha Knöbl, Tatiana C. M. Bosqueiro, Elder Sano, Gladyston C. V. Costa, Cairo Monteiro, Erick G. Dorlass, Nathalia Utecht, Guilherme P. Scagion, Carla Meneguin, Laura M. N. Silva, Maria Vitória S. Moraes, Larissa M. Bueno, Dilmara Reischak, Adriano O. T. Carrasco, Clarice W. Arns, Helena L. Ferreira, Edison L. Durigon

**Affiliations:** 1Biomedical Science Institute, University of São Paulo, São Paulo 05508-000, Brazil; lucmt@usp.br (L.M.T.); danibruna@gmail.com (D.B.L.O.); elderpereira@prefeitura.sp.gov.br (E.S.); cairomonteiro00@gmail.com (C.M.); erickgd@usp.br (E.G.D.); nathalia.utecht@gmail.com (N.U.); gui.scagion@gmail.com (G.P.S.); carla.barbosa@usp.br (C.M.); eldurigo@usp.br (E.L.D.); 2Zoonoses Surveillance Division of the Health Surveillance Coordination, Health Department of São Paulo, R. Santa Eulália, 86, São Paulo 02031-020, Brazil; janaya@prefeitura.sp.gov.br (J.A.S.); tcmoreira@prefeitura.sp.gov.br (T.C.M.B.); gccosta@prefeitura.sp.gov.br (G.C.V.C.); 3Hospital Israelita Albert Einstein, São Paulo 05652-900, Brazil; 4Avian Medicine Laboratory, Veterinary Medicine and Animal Science School, University of São Paulo, Butantã, São Paulo 05508-270, Brazil; tknobl@usp.br; 5Graduate Program in Experimental Epidemiology Applied to Zoonoses, Veterinary Medicine and Animal Science School, University of São Paulo, São Paulo 05508-270, Brazil; lauramonasc@usp.br (L.M.N.S.); mariavitoriamoraes@usp.br (M.V.S.M.); 6Department of Veterinary Medicine, FZEA-USP, University of Sao Paulo, Pirassununga 13635-900, Brazil; buenolm@usp.br; 7Federal Laboratory for Agricultural Defense in Sao Paulo (LFDA-SP), Animal Diagnostics Unit, Rua Raul Ferrari, s/n°, Campinas 13100-105, Brazil; dilmara.reischak@agricultura.gov.br; 8Department of Veterinary Medicine, State University of the Midwest, Alameda Élio Antonio Dalla Vecchia, 838, Guarapuava 85040-167, Brazil; adriano.carrasco@gmail.com; 9Department of Genetics, Evolution and Bioagents, Institute of Biology, University of Campinas—UNICAMP, P.O. Box 6109, Campinas 13083-970, Brazil; clarns@gmail.com

**Keywords:** pigeon, urban, Newcastle disease virus, avian paramyxovirus

## Abstract

Newcastle disease virus (NDV) can infect over 250 bird species with variable pathogenicity; it can also infect humans in rare cases. The present study investigated an outbreak in feral pigeons in São Paulo city, Brazil, in 2019. Affected birds displayed neurological signs, and hemorrhages were observed in different tissues. Histopathology changes with infiltration of mononuclear inflammatory cells were also found in the brain, kidney, proventriculus, heart, and spleen. NDV staining was detected by immunohistochemistry. Twenty-seven out of thirty-four tested samples (swabs and tissues) were positive for Newcastle disease virus by RT-qPCR test, targeting the M gene. One isolate, obtained from a pool of positive swab samples, was characterized by the intracerebral pathogenicity index (ICPI) and the hemagglutination inhibition (HI) tests. This isolate had an ICPI of 0.99, confirming a virulent NDV strain. The monoclonal antibody 617/161, which recognizes a distinct epitope in pigeon NDV strains, inhibited the isolate with an HI titer of 512. A complete genome of NDV was obtained using next-generation sequencing. Phylogenetic analysis based on the complete CDS F gene grouped the detected isolate with other viruses from subgenotype VI.2.1.2, class II, including one previously reported in Southern Brazil in 2014. This study reports a comprehensive characterization of the subgenotype VI.2.1.2, which seems to have been circulating in Brazilian urban areas since 2014. Due to the zoonotic risk of NDV, virus surveillance in feral pigeons should also be systematically performed in urban areas.

## 1. Introduction

Avian orthoavulavirus 1, also known as avian paramyxovirus serotype 1 (APMV-1) or Newcastle disease virus (NDV), belongs to the genus Orthoavulavirus of the family Paramyxoviridae [1]. NDV is classified into two distinct classes, class I and II, all within a single serotype [2]. Class II viruses have a great genetic diversity, with further classification into 20 genotypes, and they are found in domestic and wild birds [3]. These viruses have infected over 250 bird species with variable pathogenicity [4]. The World Organization of Animal Health (OIE) defines the virulent NDV (vNDV) isolates as those with intracerebral pathogenicity index values of over 0.7 and the F protein cleavage site containing the amino acid motif characteristic of virulence in chickens [5]. NDV outbreaks are controlled by live attenuated, inactivated, and recombinant vaccines in the poultry industry [6].

Some vNDV isolates of genotype VI, which are recognized by a unique monoclonal antibody (MAb) binding profile [7], have wild and domestic birds of the family Columbidae as their reservoir [8]. They are also known as pigeon paramyxoviruses 1 (PPMV-1) as they cause neurological disorders with high mortality rates when they infect these hosts [9]. First emerging in the Middle East in the late 1970s, PPMV-1 has been found worldwide [9,10,11,12]. Although PPMV-1 isolates meet the OIE criteria for vNDV, they have lower pathogenicity and transmissibility in chickens [13]. Some studies suggest that PPMV-1 can increase in pathogenicity in chickens after successive passages [14].

In rare cases, vaccinal and virulent NDV strains can cause clinical signs in humans, such as unilateral or bilateral conjunctivitis, lachrymation, or edema of eyelids. The infections are usually transient, and the cornea is not affected [4]. However, acute keratoconjunctivitis caused by coinfection with NDV and human adenovirus was recently reported [15]. Fatal pneumonia caused by infection with PPMV-1 in immunocompromised patients, likely infected from pigeons or doves, was also documented in the USA and the Netherlands [16,17]. Therefore, surveillance of NDV needs to be carried out in poultry farms and in urban environments. The present study reports an outbreak of feral pigeons located in Sao Paulo, one of the world’s largest cities.

## 2. Case Report

During the Brazilian winter of 2019 (July to September), over 58 feral pigeons (*Columba livia*) died, and 25 were collected in São Paulo city by the Zoonoses Surveillance Division (DVZ) of the Health Surveillance Coordination, São Paulo Municipal Health Department. Sample collection, virus characterization, and virus isolation were approved by and performed following the Institutional Animal Care and Use Committee (IACUC) (CEUA-FZEA-USP: 1355190521 and CEUA-ICB-USP113/2017).

Two out of twenty-five pigeons were selected for histopathology and sent to the avian medicine laboratory, Veterinary Medicine and Animal Science School, University of São Paulo (FMVZ-USP). Collected tissues were prepared for histopathology and immunohistochemistry (IHC), as previously described (Dimitrov et al., 2019), with few modifications. A 1:100 dilution of a primary polyclonal rabbit-derived anti-NDV La Sota strain (BS-10044R, Thermo Fisher, Waltham, MA, USA) was applied and allowed to incubate overnight at 4 °C. The primary antibody was then detected by using the EasyLink’s one anti-immunoglobulin kit with an HRP detection system (Grupo Erviegas, Indaiatuba, Brazil). DAB (Thermo Fisher, USA) served as the substrate chromogen, and hematoxylin was used as a counterstain. Pigeons showed neurological signs, including prostration, disorientation, incoordination, opisthotonos, wing paralysis, and regurgitation of liquid content. In most birds, lung congestion, diffuse hepatitis, mild necrosis, and hemorrhages in the pancreas, duodenum, and brain were present. Histopathological changes were observed in different tissues from infected pigeons. The infiltration of mononuclear cells was observed in many tissues, such as the kidney, heart, brain, intestine, and proventriculus. NDV protein antigen staining was detected in intracytoplasmic and extracellular areas of necrosis neurons in the brain and mucous membrane cells of the tertiary duct in the proventriculus (Figure 1).

Thirty-four samples (cloacal, oropharyngeal swabs, trachea, and intestine) from 23 pigeons were sent to the Clinical and Molecular Virology Laboratory, Biomedical Sciences Institute, University of São Paulo (LVCM-USP), and LFDA-SP. Cloacal swabs (n = 22), oral swabs (n = 5), large intestine (n = 4), small intestine (n = 2), and trachea (n = 1) samples were collected for screening of viral pathogens by molecular tests. All samples were extracted by the MagMax automated method (ThermoFisher) following the manufacturer’s guidelines and using the MagMax Total Nucleic acid CORE Purification kit (Thermo Fisher, Carlsbad, CA, USA). Nucleic acids were evaluated by specific real-time RT-PCR (RT-qPCR) for Alphavirus, Flavivirus, West Nile virus, Avian Influenza virus, and Newcastle disease virus, strictly following the referenced protocols [18,19,20,21,22]. The NDV M gene was detected in 27 samples from 19 pigeons, and only four tested birds were negative, comprising seven samples (Table 1). The NDV F gene was detected in 24 samples from 17 pigeons. Twenty-three samples were detected by both RT-qPCR targeting M and F genes from 16 pigeons. All samples were negative for the other tested pathogens.

Cloacal swab samples from seven pigeons out of twenty-five were collected by the Official Veterinary Service and sent to the Federal Laboratory for Agricultural Defense (LFDA-SP), an OIE reference laboratory, for molecular and biological characterization of NDV. Swab samples were pooled within up to four swabs for virus isolation due to the small volume. The obtained isolate was further tested by the hemagglutination inhibition test (HI) and the intracerebral pathogenicity test (ICPI) using the standard methodologies of OIE [5], following the ABNT NBR ISO/IEC 17025 Standard. Briefly, the HI test was performed using reference antisera produced by APHA Scientific and NVSL/APHIS/USDA with a panel of antibodies specific for avian paramyxoviruses (APMV) types -1 to -9 (except for APMV-5), and the 16 subtypes of the influenza A virus. Additionally, the isolated virus was evaluated against the monoclonal antibodies mAb 7D4 (specific for LaSota vaccine strain), mAb U85 (specific for classical strains), and mAb 617/161 (specific for pigeon strains) produced by APHA Scientific. Any HI titer greater than or equal to 16 was considered positive. The ICPI test was performed using 0.05 mL of the diluted (1:10) isolate by intracerebral route in 10 one-day-old SPF chicks. Birds were kept in isolators and monitored daily for 8 days. Birds were scored as 0 if normal, 1 if sick, or 2 if dead. The mean score per bird per observation over the 8 days was calculated to obtain the ICPI value. The animal experiment and virus isolation were also approved by and performed per IACUC in animal biosecurity level 3 enhanced (ABSL-3E) facilities at LFDA-SP. One isolate was obtained from the pool with samples LF2, LF3, LF5, and LF7. It was inhibited by monoclonal Ab 617/161 with higher titers compared to the other mAbs. Hemagglutination inhibition titers of the isolated virus with the mAb 7D4, U85, and 617/161 were: 8 (uncomplete), 32 (uncomplete), and 512, respectively. The ICPI index was 0.99, which is considered virulent.

Total nucleic acid from the oral swab sample CCZ002, named here as NDV/pigeon/SP-Brazil/CCZ002/2019, which had the lowest Ct by RT-qPCR targeting the M gene (Ct = 15.1), was reverse-transcribed using SuperScript VILO reverse transcriptase (Thermo Fisher). A second-strand extension was performed using Invitrogen Second Strand cDNA Synthesis Kit (Thermo Fisher). Double-stranded cDNA was purified with Agencourt AMPure XP bead purification (Beckman Coulter, Brea, CA, USA; 1.8× ratio). Sample dilution and library construction were performed according to the manufacturer’s instructions for the Ion Xpress™ Plus Fragment Library Kit (Thermo Fisher). Individually barcoded libraries (Ion Express Barcode Adapters—Thermo Fisher) were linked with Ion Amplicon Adapter Ligation after AMPure XP bead cleanup (1.8× ratio) and before AMPure XP bead cleanup (1.0× ratio). The products were size-selected using E-Gel SizeSelect II gels (Thermo Fisher). Amplified libraries were purified (AMPure ×P bead cleanup 1.0× ratio) and quantified before pooling using the Ion Library TaqMan Quantitation Kit (Thermo Fisher). The Ion Torrent S5 libraries were prepared using the “Ion 510™ & Ion 520™ & Ion 530™ kit” of the Ion Chef Kit for 400 base-read libraries and sequenced on the Ion Torrent S5 using an Ion 530 semi-conductor sequencing chip (Thermo Fisher Scientific, Carlsbad, CA, USA). Obtained raw reads were mapped to reference (KT163264) using Geneious Prime. A consensus sequence with 15,192 bp length had a coverage of 100% with 111,208 sequencing reads and a mean depth of 2683.5 times. The obtained complete genome of the sample NDV/pigeon/SP-Brazil/CCZ002/2019 was submitted to GenBank under access number MZ458602.

Phylogenetic analysis was performed using the Maximum Likelihood method based on the complete F NDV gene on the General Time Reversible model [23]. Initial tree(s) for the heuristic search were obtained automatically by applying Neighbor-Join and BioNJ algorithms to a matrix of pairwise distances estimated using the Maximum Composite Likelihood (MCL) approach and then selecting the topology with a superior log-likelihood value. A discrete Gamma distribution was used to model evolutionary rate differences among sites (5 categories (+G, parameter = 0.6753)). The tree was drawn to scale, with branch lengths measured in the number of substitutions per site. The analysis involved 67 nucleotide sequences. Codon positions included were 1st+2nd+3rd+Noncoding. All positions containing gaps and missing data were eliminated. There were a total of 1653 positions in the final dataset. Evolutionary analyses were conducted in MEGA7 [24]. The evolutionary divergence between sequences was conducted using the Maximum Composite Likelihood model [25]. The obtained sequence was grouped with sequences from the genotype VI.2.1.2 detected in Africa (Nigeria and Kenya) and South America (Argentina and Brazil) (Figure 2).

The genetic identities of sequences from the genotype VI.2.1.2 varied from 91.9% to 97.9% (Table 2). The obtained sequence in this study had the highest nucleotide identity (97.9%), with the sequence from Brazil detected in 2014. Our isolate had three basic amino acids (two arginines and one lysine) between residues 113 and 116 in the C-terminus of the F2 protein and phenylalanine at residue 117 in the N-terminus of the F1 protein (113RQKR↓F117). Such cleavage sites are specific for virulent viruses based on criteria utilized by OIE to assess the virulence of NDV isolates [5].

## 3. Discussion

The feral pigeons of the present study were found in the metropolitan region of the largest urban center in Latin America. The authorities were notified, and the mitigation actions protocol was initiated. As it is a large urban center, there are no breeding creations in the nearby area. The virus is a serious threat to wild and domestic pigeons with high mortality rates and neurological disorders [4]. In the present study, birds displayed neurological clinical signs with necrosis in multiple organs, as well as histopathological changes.

PPMV-1 is endemic in many countries, and different data support that they do not pose a high threat to the poultry industry [13]. In the current outbreak, 10 chickens (*Gallus gallus*), which were located close to the sick pigeons, did not show clinical signs (data not shown). Nevertheless, infections with other vNDV genotypes more chicken-adapted have been reported in wild pigeons [26,27]. The last NDV genotype classification split the viruses from genotype VI into a new genotype (XXI) [3]. The genotype XXI can cause clinical signs in chickens without prior adaptation [28]. The diagnostic differentiation using the monoclonal antibodies remains important to distinguish the PPMV-1, especially the subgenotype VI, from other genotypes, but this is not always accessible [7]. Here, a PPMV-1 isolate from subgenotype VI.2.1.2 could be characterized using this panel. The isolate was also classified as a virulent strain by the ICPI test that has an ICPI value greater than 0.7.

The NDV subgenotype VI.2.1.2 was reported in Argentina, Brazil, Nigeria, and Kenya. This subgenotype was previously detected in feral pigeons from a public square in Porto Alegre, the Rio Grande do Sul state, southern Brazil, in 2014 [29]. Although the sequences from Brazil clustered with sequences from Nigeria, Kenya, and Argentina, they have genetic distances greater than 0.05 with a high bootstrap, which suggests that they belong to another subgenotype. However, a new subgenotype can only be suggested with sequences from at least three distinct epidemiological events [3]. The lack of sequences indicates that the notification of diseases in non-commercial birds is limited in Latin American countries, including Brazil.

The present study is the first comprehensive report of the subgenotype V.2.1.2 of NDV detected with biological and molecular characterization. The NDV subgenotype V.2.1.2, initially identified in another urban area in 2014 based on a partial F gene sequence, seemed to continuously circulate in Brazil as it appeared five years later 1100 km away from its prior identification. Feral pigeon populations with zoonotic pathogens in urban areas can pose a low risk to healthy humans, but this risk can be increased up to 1000 times in immunocompromised patients [30]. PPMV-1 remains zoonotic with generally mild infection in humans, usually conjunctivitis, although two fatal pneumonia cases were already reported in immunocompromised persons [4,16,17]. Precautious measures should be followed when manipulating wild or domestic animals. According to OIE, 75% of emerging human pathogens are of animal origin. Thus, the maintenance of active surveillance and the increase in biosafety measures to control pigeon populations away in urban areas and poultry farms are essential.

## Figures and Tables

**Figure 1 viruses-13-02446-f001:**
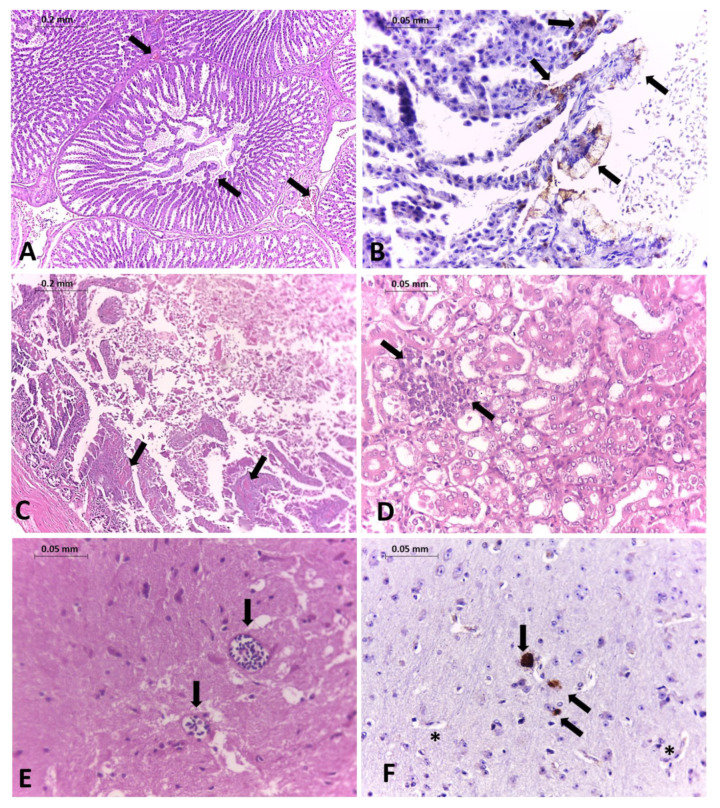
Histopathology changes and immunohistochemistry (IHC) in tissues from a naturally infected pigeon. HE staining. (**A**) Proventriculus with hemorrhages and edema (arrows); objective magnification 10×. (**B**) NDV staining in mucous membrane cells of tertiary duct in the proventriculus (arrows); objective magnification 40×. (**C**) Intestine with necrosis; objective magnification 40×. (**D**) Kidney infiltration of mononuclear inflammatory cells with necrosis of tubular epithelial cells (arrows); objective magnification 40×. (**E**) Multifocal lymphoplasmacytic perivascular cuffs in the brain (arrows); objective magnification 40×. (**F**) NDV protein antigen staining was detected in intracytoplasmic (arrows) and extracellular areas of necrosis neurons (asterisk) of the brain; objective magnification 40×.

**Figure 2 viruses-13-02446-f002:**
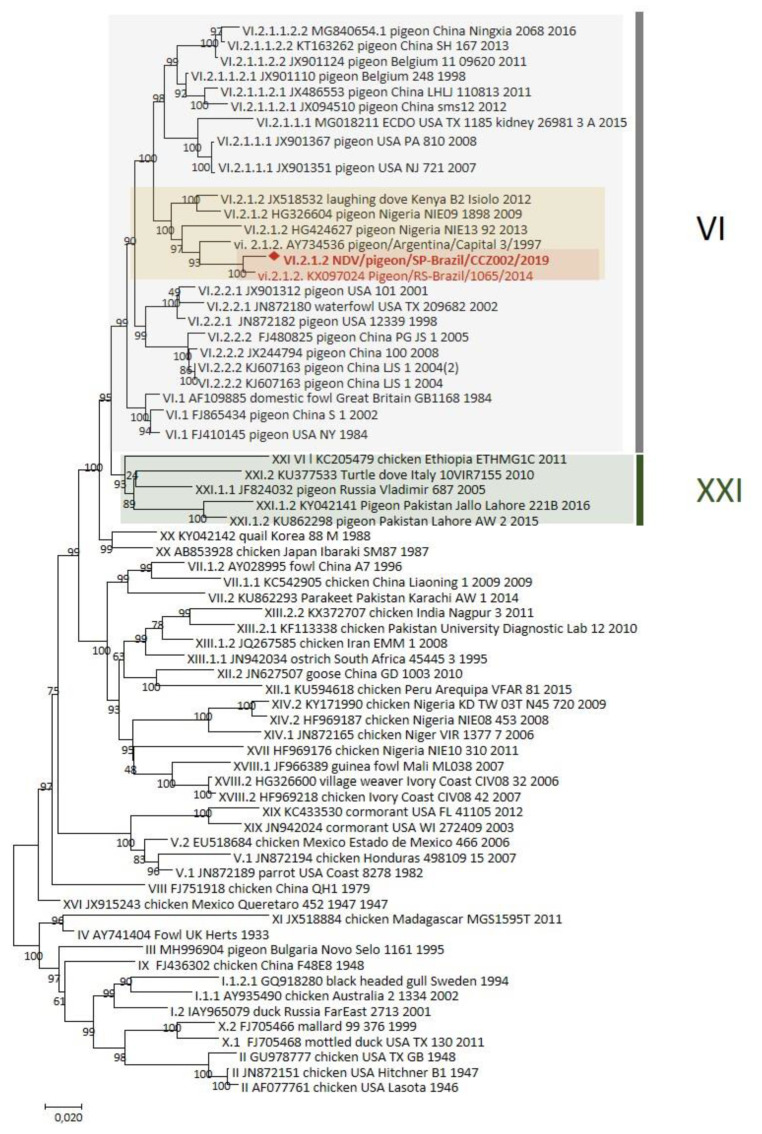
The evolutionary history was inferred by using the Maximum Likelihood method based on complete F NDV gene from the General Time Reversible model [1]. The tree with the highest log likelihood (-20110.81) is shown. The percentage of trees in which the associated taxa clustered together is shown next to the branches. Brazilian samples are highlighted in red, and sequences from the subgenotype VI.2.1.2 are highlighted in yellow. The sequence NDV/pigeon/SP-Brazil/CCZ002/2019 is also highlighted with a red diamond.

**Table 1 viruses-13-02446-t001:** Description of collected samples during the NDV outbreak from July to September 2019 in Sao Paulo city, Brazil.

Sample ID	Type of Sample	Collection Date	Sent to	Ct Values NDV M Gene	Ct Values NDV F Gene
CCZ001	CS	24/07/2019	LVCM-USP	26.5	24.7
CCZ002	CS	24/07/2019	LVCM-USP	18.4	29.4
CCZ002	OS	24/07/2019	LVCM-USP	15.1	35.3
CCZ003	CS	31/07/2019	LVCM-USP	19.5	27.1
CCZ004	CS	05/08/2019	LVCM-USP	21.6	Negative
CCZ005	CS	05/08/2019	LVCM-USP	22.1	30.9
CCZ006	CS	05/08/2019	LVCM-USP	35.1	Negative
CCZ007	CS	06/08/2019	LVCM-USP	20.5	28.8
CCZ0030	LI	08/08/2019	FMVZ/LVCM-USP	23.9	29.2
CCZ0031	CS	12/08/2019	FMVZ/LVCM-USP	24.7	30.4
CCZ0031	LI	12/08/2019	FMVZ/LVCM-USP	27.4	39.9
CCZ0031	OS	12/08/2019	FMVZ/LVCM-USP	28.6	34.5
CCZ0031	TR	12/08/2019	FMVZ/LVCM-USP	36.8	Negative
CCZ0032	CS	12/08/2019	FMVZ/LVCM-USP	26.7	32.6
CCZ0032	OS	12/08/2019	FMVZ/LVCM-USP	Negative	Negative
CCZ0033	CS	12/08/2019	FMVZ/LVCM-USP	22.6	28.2
CCZ0033	OS	12/08/2019	FMVZ/LVCM-USP	26.8	32.1
CCZ0034	CS	13/08/2019	FMVZ/LVCM-USP	Negative	Negative
CCZ0034	OS	13/08/2019	FMVZ/LVCM-USP	Negative	Negative
CCZ0035	CS	14/08/2019	FMVZ/LVCM-USP	23.4	28.8
CCZ0035	SI	14/08/2019	FMVZ/LVCM-USP	28.9	33.9
CCZ0038	CS	17/08/2019	LVCM-USP	Negative	Negative
CCZ0038	LI	17/08/2019	LVCM-USP	Negative	Negative
CCZ0041	CS	20/08/2019	LVCM-USP	27.2	32.2
CCZ0041	SI	20/08/2019	LVCM-USP	27.5	32.6
CCZ0043	CS	20/08/2019	LVCM-USP	22.0	27.9
CCZ0043	LI	20/08/2019	LVCM-USP	22.1	27.1
LF1	CS	10/09/2019	LFDA-SP	Negative	35.6
LF2	CS	10/09/2019	LFDA-SP	19.5	32.1
LF3	CS	10/09/2019	LFDA-SP	22.9	34.5
LF4	CS	10/09/2019	LFDA-SP	Negative	Negative
LF5	CS	10/09/2019	LFDA-SP	24.2	34.8
LF6	CS	10/09/2019	LFDA-SP	36.9	Negative
LF7	CS	10/09/2019	LFDA-SP	23.1	35

OS = oral swab; CS = cloacal swab; LI = large intestine; TR = trachea; SM = small intestine, BR = brain. FMVZ: Faculty of Veterinary Medicine and Animal Health (USP), LVCM: Laboratory of Clinical and Molecular Virology (USP); LFDA-SP: Federal Laboratory for Agricultural Defense in Sao Paulo; LMVPA: Laboratory of Applied Preventive Veterinary Medicine (USP).

**Table 2 viruses-13-02446-t002:** Genetic distances of viruses from subgenotype VI.2.1.2.

ID	Strain Name	ID					
		1	2	3	4	5	6
1	NDV/pigeon/SP-Brazil/CCZ002/2019						
2	KX097024_Pigeon/RS-Brazil/1065/2014	97.9					
3	AY734536_pigeon/Argentina/Capital_3/1997	94.3	94.9				
4	JX518532_laughingdove/Kenya/B2/Isiolo/2012	92.0	92.7	94.3			
5	HG424627 pigeon/Nigeria/NIE13/92/2013	92.7	93.4	93.7	93.8		
6	HG326604 pigeon/Nigeria/NIE09/1898/2009	91.9	92.6	93.6	97.3	93.7	

## Data Availability

The complete genome of the sample NDV/pigeon/SP-Brazil/CCZ002/2019 described in this work was submitted to GenBank under access number MZ458602.

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
