# Peer review of "An Outbreak in Pigeons Caused by the Subgenotype VI.2.1.2 of Newcastle Disease Virus in Brazil"

_viruses, 2021, doi:10.3390/v13122446_

Round 1

Reviewer 1 Report

Authors thoroughly characterized PPMV-1 from an outbreak in pigeons in Sau Paulo city, Brazil in 2019. Of the sampled birds, 21 were positive for PPMV-1 and one isolate was chosen for further characterization. The sequence of this isolate was characterized along with additional isolates from the region. The study would be stronger if more than one isolate from the out break was characterized, however it is unlikely multiple linages would have been discovered within the outbreak. The data presented is not a new finding that will make a large impact, however the case study is clearly described and adds to the knowledge base. I recommend minor grammatical editing prior to publication. 

Author Response

Thank you for your comments. The English language was revised by a native speaker in English in this revised version. The certificate was attached in the submission.

Reviewer 2 Report

The manuscript entitled " An outbreak in pigeons caused by the subgenotype VI.2.1.2 of Newcastle disease virus in Brazil" is a well written study that reports a comprehensive characterization of the NDV subgenotype VI.2.1.2 circulated in Sao Paulo city in Brazil. However, here are several issues that need to be addressed.

There are inaccurate or inconsistent descriptions in the text:

  1. In the abstract (line 32), the authors indicated that 21 out of 25 samples (swabs and tissues) were positive for Newcastle disease virus, while 34 samples were investigated as described in the main text (line 106).
  2. In the abstract (lines 33-36), the authors indicated that one virus was selected for virus isolation and further characterization, which was confusing; and the isolate have an ICPI of 0.99, …. However, in the main text (line 118), the authors indicated 7 samples were sent to HI and ICPI tests….
  3. In the abstract (lines 38-40), the authors indicated that the isolate was from subgenotype VI.1.2.1 based on the phylogenetic analysis of the complete CDS of NDV F gene, but as described in the title and main text, it was from subgenotype VI.2.1.2.
  4. In lines 77-78, “Fifteen pigeons were selected for histopathological and bacteriological examination …”, but in lines 106-107, “Thirty four samples…from 23 pigeons were sent to …”
  5. In line 132, “Total nucleic acid from Pigeon #2, which had the lowest Ct by RT-qPCR, was reverse-transcribed…”, the lowest Ct was for the NDV M gene but not for the NDV F gene. Was the isolate from Pigeon #2?

Some other issues that need to be addressed:

  1. Please rephrase the sentence in lines 34-36 (“The hemagglutination inhibition assay identified…from genotype VI, …”) in abstract, which is confusing.
  2. As described by the authors, the NDV M gene was detected in 27 samples from 18 pigeons and the NDV F gene was detected in 24 samples from 17 pigeons, how many pigeons were positive for both NDV F and M genes?
  3. What is the cut-off value for the RT-qPCR experiment in this study?
  4. How did authors conduct the Intra-cerebral pathogenicity index (ICPI) tests in this study? Please briefly describe it.
  5. Please indicate the scale bar and highlight histopathology changes in figure 2.
  6. In line 130, the authors indicated that the isolated virus was inhibited by monoclonal Ab 617/161. Please present the inhibition titers and cut-off values in the manuscript.
  7. Was the sequence of NDV/pigeon/SP-Bra-177 zil/CCZ002/2019 submitted to GenBank? Please include the reference number in the manuscript.
  8. Typos: “F NDV gene” in line 116.

Author Response

Reviewer #2

The manuscript entitled " An outbreak in pigeons caused by the subgenotype VI.2.1.2 of Newcastle disease virus in Brazil" is a well written study that reports a comprehensive characterization of the NDV subgenotype VI.2.1.2 circulated in Sao Paulo city in Brazil. However, here are several issues that need to be addressed.

Thank you for your comments.

There are inaccurate or inconsistent descriptions in the text:

  1. In the abstract (line 32), the authors indicated that 21 out of 25 samples (swabs and tissues) were positive for Newcastle disease virus, while 34 samples were investigated as described in the main text (line 106).

Lines 32-33: The sentence was revised. The total number of analyzed samples were 34 samples.

  1. In the abstract (lines 33-36), the authors indicated that one virus was selected for virus isolation and further characterization, which was confusing; and the isolate have an ICPI of 0.99, …. However, in the main text (line 118), the authors indicated 7 samples were sent to HI and ICPI tests….

Lines 34 to 37: The text was revised accordingly: “One isolate, obtained from a pool of positive swab samples, was characterized by the intracerebral pathogenicity index (ICPI) and the hemagglutination inhibition (HI) tests. This isolate had an ICPI of 0.99, confirming a virulent NDV strain. The monoclonal antibody 617/161, which is specific to recognize pigeon strains, inhibited the isolate with HI titer of 512”

The sentence was revised

  1. In the abstract (lines 38-40), the authors indicated that the isolate was from subgenotype VI.1.2.1 based on the phylogenetic analysis of the complete CDS of NDV F gene, but as described in the title and main text, it was from subgenotype VI.2.1.2.

Lines 39 and 41: The subgenotype VI.2.1.2 was corrected in the abstract.

  1. In lines 77-78, “Fifteen pigeons were selected for histopathological and bacteriological examination …”, but in lines 106-107, “Thirty four samples…from 23 pigeons were sent to …”

Only two pigeons were sent for histopathological examination. The bacteriological examination was removed from the text to be more accurate as those samples were only tested for bacteria and those samples were not tested for NDV

  1. In line 132, “Total nucleic acid from Pigeon #2, which had the lowest Ct by RT-qPCR, was reverse-transcribed…”, the lowest Ct was for the NDV M gene but not for the NDV F gene. Was the isolate from Pigeon #2?

Sample CCZ002 was identified accordingly in the text and table 1:

The sentence was revised.

Line 142-144: “Total nucleic acid from the sample CCZ002, named here in as NDV/pigeon/SP-Brazil/CCZ002/2019, which had the lowest Ct by RT-qPCR targeting the M gene (Ct=15.1), was reverse-transcribed using SuperScript VILO reverse transcriptase (Thermo Fisher).”

Some other issues that need to be addressed:

  1. Please rephrase the sentence in lines 34-36 (“The hemagglutination inhibition assay identified…from genotype VI, …”) in abstract, which is confusing.

The text was modified as follows:

Line 36 -37: “The monoclonal antibody 617/161, which is specific to recognize pigeon strains, inhibited the isolate with HI titer of 512”

  1. As described by the authors, the NDV M gene was detected in 27 samples from 18 pigeons and the NDV F gene was detected in 24 samples from 17 pigeons, how many pigeons were positive for both NDV F and M genes?

A sentence was added to clarify this information.  was modified as follows: Lines 116-117: Twenty-three samples were detected by both RT-qPCR targeting M and F genes.

  1. What is the cut-off value for the RT-qPCR experiment in this study?

A sentence was added to clarify the information.

Lines 113-114: Any sample with a Ct value below the cutoff of 40 cycles was considered as positive.

  1. How did authors conduct the Intra-cerebral pathogenicity index (ICPI) tests in this study? Please briefly describe it.

Lines 130-134: The information was included in the text: “The ICPI test was performed using 0.05 of the diluted (1:10) isolate by intracerebral route in ten one-day-old SPF chicks. Birds were kept in isolators and daily monitored during 8 days. Birds were scored as 0 if normal, 1 if sick or 2 if dead. The mean score per bird per observation over the 8 day-period was calculated to obtain the ICPI value.”

  1. Please indicate the scale bar and highlight histopathology changes in figure 2.

Figure 2 and lines 98-103: Information was added.

  1. In line 130, the authors indicated that the isolated virus was inhibited by monoclonal Ab 617/161. Please present the inhibition titers and cut-off values in the manuscript.

The information was included:

Lines 131-134: Any HI titer greater than or equal 16 was considered as positive.

Lines 136-141: The isolate was inhibited by monoclonal Ab 617/161 with higher titers compared to the other mAb. Hemagglutination inhibition titers of the isolated virus with the mAb 7D4, U85, and 617/161 were: 8 (incomplete), 32 (incomplete), and 512, respectively. The ICPI index was 0.99, which is considered virulent”

  1. Was the sequence of NDV/pigeon/SP-Brazil/CCZ002/2019 submitted to GenBank? Please include the reference number in the manuscript.

The sentence was submitted and the GenBank number was described in lines 263-264. An additional sentence was added in the main text to clarify this (lines 160-162): “The obtained complete genome of the sample NDV/pigeon/SP-Brazil/CCZ002/2019 was submitted to GenBank under access number MZ458602.”

  1. Typos: “F NDV gene” in line 116.

The typo was corrected

Reviewer 3 Report

Regarding a case report entitled "An outbreak in pigeons caused by the subgenotype VI.2.1.2 of 2 Newcastle disease virus in Brazil", the authors aimed to investigate an outbreak in feral pigeons in 28 São Paulo city, Brazil, in 2019.

The case report is interesting and well written. Therefore, I suggest its acceptance after an English language editing.

Author Response

Reviewer #3

Regarding a case report entitled "An outbreak in pigeons caused by the subgenotype VI.2.1.2 of 2 Newcastle disease virus in Brazil", the authors aimed to investigate an outbreak in feral pigeons in 28 São Paulo city, Brazil, in 2019.

The case report is interesting and well written. Therefore, I suggest its acceptance after an English language editing.

The English language was revised by a native speaker in English in this revised version. The certificate was attached in the submission.